# Disparities in Metabolic Conditions and Cancer Characteristics among Hispanic Women with Breast Cancer: A Multi-Institutional Study

**DOI:** 10.3390/cancers14143411

**Published:** 2022-07-14

**Authors:** Iktej S. Jabbal, Alok Dwivedi, Nadeem Bilani, Barbara Dominguez, Gehan Botrus, Zeina Nahleh

**Affiliations:** 1Department of Hematology & Oncology, Maroone Cancer Center, Cleveland Clinic, Weston, FL 33331, USA; domingb3@ccf.org; 2Department of Molecular and Translational Medicine, Texas Tech University Health Sciences Center, El Paso, TX 79905, USA; alok.dwivedi@ttuhsc.edu; 3Department of Internal Medicine, Icahn School of Medicine at Mount Sinai, New York, NY 10029, USA; nadeem.bilani@mountsinai.org; 4Department of Hematology & Oncology, Mayo Clinic in Arizona, Phoenix, AZ 85054, USA; botrus.gehan@mayo.edu

**Keywords:** breast cancer, Hispanic ethnicity, metabolic abnormalities

## Abstract

**Simple Summary:**

This multi-institutional analysis suggests associations of common metabolic conditions with ethnicity among Hispanic patients with breast cancer. Hispanic individuals with breast cancer are diverse and have been historically lumped under one category in research protocols without distinction or reference to their country of origin. This study highlights differences in tumor characteristics and their associations with metabolic conditions among the various Hispanic patients with breast cancer based on their ethnic origins, which should be considered when referencing race and ethnicity. This study supports a more focused approach to addressing obesity and other metabolic conditions in patients with breast cancer within the Hispanic population. In addition, the authors aim to increase awareness regarding the prevalence of common metabolic conditions in the Hispanic population and recommend measures to improve overall health and breast cancer care, including prioritizing lifestyle modifications for Hispanics and other minorities.

**Abstract:**

While the associations of common metabolic conditions with ethnicity have been previously described, disparity among Hispanic individuals based on country of origin is understudied. This multi-institutional analysis explored the prevalence of metabolic conditions and their association with cancer subtypes among Mexican and non-Mexican Hispanics. After IRB approval, we conducted a cross-sectional study at two academic medical centers with a significant Hispanic patient population (Texas Tech University Health Sciences Center, El Paso, TX (TTUHSC-EP) and Cleveland Clinic Florida in Weston, FL (CCF)). A total of *n* = 1020 self-identified Hispanic patients with breast cancer consecutively diagnosed between 2005 and 2014 were selected from the two institutional databases. Comparisons between Mexican and Non-Mexican Hispanics revealed variations in tumor types and metabolic conditions. Mexican Hispanics were found to have a higher prevalence of diabetes mellitus (27.8% vs. 14.2%, *p* < 0.001), obesity (51.0% vs. 32.5%, *p* < 0.001), and ductal carcinoma type (86.6 vs. 73.4%, *p* < 0.001). On the other hand, hormone-receptor-positive breast cancer was more common in non-Mexicans, while Mexicans had more triple-negative breast cancer, especially in premenopausal women. In addition to highlighting these variations among Hispanic patients with breast cancer, this study supports a more focused approach to addressing obesity and other metabolic conditions prevalent in the Hispanic population with breast cancer. Moreover, Hispanic individuals with breast cancer are diverse and should not be lumped under one category without reference to their country of origin regarding the impact of race and ethnicity.

## 1. Introduction

Breast cancer is the second-most common cause of cancer-related death in women in the United States [1]. The age-adjusted incidence is highest among non-Hispanic White women, followed by African American and Hispanic women [2,3]. Individuals with breast cancer diagnosed with metabolic abnormalities such as diabetes mellitus (DM), dyslipidemia, hypertension (HTN), and obesity have been reported to have reduced overall survival [4]. However, it remains unclear whether the complex etiology of these metabolic abnormalities leads to an increased risk for breast cancer or whether it affects the severity of disease presentation overall. Furthermore, the potential association of metabolic characteristics with tumor subtype is under-studied. Nonetheless, their presence tends to increase the complexity of the clinical decision-making process due to their significant impact on treatment outcomes.

While it has been previously reported that metabolic abnormalities are relatively common in the Hispanic population and that Hispanic patients should not be clubbed together as a single entity [5], differences amongst the Hispanic population based on their country of origin are yet to be explored. Understanding these metabolic and clinical associations would provide much-needed guidance for developing more tailored preventive and treatment strategies.

This analysis aimed to explore the prevalence of metabolic abnormalities in Mexican and non-Mexican Hispanic women with breast cancer. In addition, the potential association of comorbid factors and ethnicity with the various breast cancer subtypes was assessed.

## 2. Materials and Methods

### 2.1. Study Population

After obtaining the Institutional Review Board (IRB) approval, a retrospective analysis was conducted at two tertiary medical centers (Texas Tech University of Health Sciences Center, El Paso, TX (TTUHSC-EP) and Cleveland Clinic Florida in Weston, FL (CCF)). Self-identified Hispanic women of any ethnic origin diagnosed with primary breast cancer between 2005 and 2014 were identified for inclusion in this analysis. Unidentified Hispanic ethnicity and missing data on tumor subtypes and metastatic cases were excluded.

### 2.2. Patient Characteristics

The following characteristics and parameters were collected in this study for the analysis: age, menopausal status (premenopausal as age < 50 years and postmenopausal as age ≥ 50 years), Hispanic origin (Mexican, Puerto Rican, Cuban, South/Central American, Spanish not otherwise specified (NOS)/Hispanic NOS, Dominican Republic), body mass index (BMI, kg/m^2^), breast cancer diagnosis (invasive ductal carcinoma, IDC; invasive lobular carcinoma, ILC; mixed; or others), stage of the tumor at diagnosis (I, II, III, IV), type of surgery (lumpectomy, mastectomy), comorbidities including diabetes mellitus (DM), dyslipidemia, hypertension (HTN), obesity (defined using body mass index: BMI ≥ 30 kg/m^2^), and coronary artery disease (CAD), as well as tumor subtypes including estrogen receptor (ER), progesterone receptor (PR), and human epidermal receptor 2 neu (HER2) status.

### 2.3. Outcomes and Comparative Group

The primary outcome variables were classified by breast cancer subtypes, including hormone receptor (HR) positive (ER+ or PR+), HER2+, or triple-negative breast cancer (TNBC if ER− and PR− and HER2−). The distribution of metabolic comorbidities was defined in four ways: (1) the presence of at least one comorbidity, (2) the number of comorbidities, (3) metabolic syndrome (considered with the presence of at least three metabolic comorbidities), or (4) individual comorbidities. The comparative group was of Hispanic origin (Mexican versus non-Mexican).

### 2.4. Statistical Considerations

The quantitative variables were described using mean and standard deviation (SD), while categorical data were described using frequency and proportion. All the proportions were computed after excluding missing data. All patient characteristics were compared between Mexican and non-Mexican Hispanics using either an unpaired t-test or chi-square test. Similarly, patient characteristics were also compared by different Mexican origins using either one-way analysis of variance (ANOVA) or the chi-square test. The adjusted factors associated with Mexican Hispanics compared to non-Mexican Hispanics were observed using multivariable logistic regression analysis. Results were summarized with an odds ratio (OR) with a 95% confidence interval (CI). The adjusted association of Hispanic origin with each tumor subtype was evaluated using multiple Poisson regression with robust variance analyses to obtain a prevalence ratio (PR) [6]. The interaction of Hispanic heritage with other characteristics was explored by including a product term in the multivariable analysis as suggested [7]. Multivariable models were developed for each primary outcome restricted to premenopausal women only in the presence of a solid and consistent interaction of Hispanic origin with menopausal status. The associations were further validated by considering different forms of metabolic comorbidities in multivariable models and assessed whether any specific form of metabolic comorbidities was associated with tumor subtypes using multivariable Poisson regression models. All considered variables were adjusted in multivariable models. Furthermore, the association of clinical characteristics with the metabolic condition within each ethnic cohort was also determined using multiple Poisson regression models. The results of the Poisson regression analysis were presented using prevalence ratio (PR) along with 95% CI and *p*-value. We followed the statistical analysis reporting recommendations [7]. A *p*-value < 0.05 was considered a statistically significant finding. All statistical analyses were carried out using STATA 17.

## 3. Results

### 3.1. Overall Sample Characteristics

*N* = 1020 Hispanic patients (*n* = 849 at TTUHSC-EP and *n* = 171 at CCF) were included in this analysis. The average age was 56 years, with a mean BMI of 30.4 kg/m^2^. Most patients had ductal carcinoma (84.4%), 48.7% had early-stage tumors, and almost half of the population received a lumpectomy for their primary tumors. Most patients had HR+ and HER 2 negative breast cancers, with ER+ (69.9%), followed by PR+ (59.2%). In addition, most patients had at least one associated comorbidity (72.7%), with obesity being the most common one (47.9%). Most of the patients in our analysis were of the Mexican ethnicity (83.4%), followed by those categorized as Spanish NOS/Hispanic NOS (12.6%) (Table 1).

### 3.2. Differences in Patient Characteristics by Hispanic Ethnicities

Table 1 also shows the comparisons of patient characteristics between Mexican Hispanics and non-Mexican Hispanic ethnicities. Mexican Hispanics had a higher average BMI than non-Mexican Hispanics (30.9 vs. 28.0, *p* < 0.001), a higher prevalence of the ductal carcinoma breast cancer subtype (86.6% vs. 73.4%, *p* < 0.001), and a higher number of received lumpectomy as the surgical option (54% vs. 32%, *p* < 0.001) compared to the others. Moreover, a higher proportion of Mexican Hispanics had DM (27.8% vs. 14.2%, *p* < 0.001) and obesity (51% vs. 32.5%, *p* < 0.001). In contrast, the majority of non-Mexican Hispanics were observed to have hyperlipidemia (50.9% vs. 24.9%) and the HR+ (especially ER+) tumor subtype (76.3% vs. 67.9%, *p* = 0.03). In the adjusted analysis, these factors remained significantly associated with Mexican Hispanics (Appendix A).

As seen in Appendix A, which included only the premenopausal subgroup, Mexican Hispanics were observed to have a higher mean BMI (30.0 vs. 28.0, *p* = 0.018). A higher proportion of patients were diagnosed with ductal carcinoma (91.1% vs. 75.4%, *p* = 0.002) without any statistically significant differences in the stage of tumors at diagnosis. Regarding the type of surgery, a higher proportion of Mexican Hispanics had undergone lumpectomy (51.1%) compared to the non-Mexican Hispanic population (40.0%) for their tumors. Lastly, both hormonal subtypes (ER and PR) were more frequently observed in non-Mexican Hispanics, while triple-negative was common in the Mexican Hispanics (27.2% versus 12.3%). A higher proportion of non-Mexican Hispanics (33.8% vs. 12.5%, *p* < 0.001) had hyperlipidemia. The mean BMI was the highest in individuals of Puerto Rican ethnicity (34.0), followed by those of Mexican ethnicity (30.9). Ductal histology was the most common type of cancer across all Hispanic races. All three patients of the Dominican Republican ethnicity were seen to be hyperlipidemic, followed by patients of Puerto Rican ethnicity (80%) and South/Central American nationality (61.9%) (Appendix A).

### 3.3. Adjusted Association of Mexican Hispanics Ethnicity with Tumor Subtypes Compared to Non-Mexcian Hispanics

The association of Hispanic origin with breast cancer subtypes is shown in Table 2. The prevalence of ER-positive tumors was 12% (PR = 1.12, 95% CI: 1.01–1.25, *p* = 0.036) higher in non-Mexican Hispanics than in Mexican Hispanics. Tumors with lobular carcinoma (PR = 1.33, 95% CI: 1.22–1.46, *p* < 0.001) and early-stage cancer (PR = 1.16, *p* = 0.01) were associated with a higher prevalence of ER positivity. Like ER, the PR tumor subtype was frequently associated with HTN (PR = 1.13, 95% CI: 1.01–1.27, *p* = 0.036) and lobular carcinoma (PR = 1.35, 95% CI: 1.17–1.55, *p* < 0.001) and less likely to be associated with being postmenopausal (PR = 0.90, 95% CI: 0.80–1.00, *p* = 0.056). Overall, HR positivity was associated more with the non-Mexican Hispanic subtype (PR = 1.13, 95% CI: 1.01–1.25, *p* = 0.028), HTN (PR = 1.13, 95% CI: 1.03–1.24, *p* = 0.008), and lobular carcinoma (PR = 1.28, 95% CI: 1.16–1.41, *p* < 0.001).

In premenopausal women (Appendix A), patients of non-Mexican Hispanic origin (PR = 1.37, 95% CI: 1.18–1.59, *p* < 0.001), lobular (PR = 1.37, 95% CI: 1.14–1.65, *p* = 0.001), or mixed carcinoma (PR = 1.29, 95% CI: 1.13–1.47, *p* < 0.001) types and DM (PR = 1.20, 95% CI: 1.00–1.45, *p* = 0.052) were associated with increased prevalence of ER tumor. In addition, premenopausal women had a higher prevalence of the PR subtype when they were of a non-Mexican Hispanic origin (PR = 1.38, 95% CI: 1.14–1.67, *p* = 0.001) and had lobular (PR = 1.47, 95% CI: 1.19–1.81, *p* < 0.001) or mixed carcinoma (PR = 1.40, 95% CI: 1.16–1.69, *p* < 0.001) subtypes. Overall, the HR subtype had a higher association with women of non-Mexican Hispanic origin (PR = 1.39, 95% CI: 1.20–1.61, *p* < 0.001) and having HTN (PR = 1.21, 95% CI: 1.04–1.40, *p* = 0.013). In contrast, the triple-negative subtype was more likely to be associated with Mexican Hispanic origin (PR = 0.43, 95% CI: 0.21–0.87, *p* = 0.019).

### 3.4. Adjusted Association of Comorbidities with Tumor Subtypes

As seen in Table 3, patients with metabolic syndrome (PR = 1.14, 95% CI: 1.04–1.25, *p* = 0.003) or patients with all four included metabolic abnormalities (PR = 1.23, 95% CI: 1.08–1.40, *p* = 0.002) had a strong association with increased prevalence of ER-positive breast cancer, or PR-positive breast cancer (PR = 1.23, 95% CI: 1.1–1.39, *p* < 0.001). In separate models, metabolic syndrome and increased metabolic abnormalities were similarly associated with an increased prevalence of HR-positive breast cancer (versus other types).

When taken separately, metabolic syndrome was associated with ER (PR = 1.12, 95% CI: 1.02–1.24, *p* = 0.023), PR (PR = 1.21, 95% CI: 1.06–1.37, *p* = 0.004), or HR positivity (PR = 1.13, 95% CI: 1.02–1.25, *p* = 0.02) in Mexican Hispanics (Appendix A). In addition, lobular carcinoma was associated with ER (PR = 1.45, 95% CI: 1.34–1.57, *p* = 0.00), PR (PR = 1.50, 95% CI: 1.31–1.73, *p* = 0.00), or HR positivity (PR = 1.38, 95% CI: 1.25–1.52, *p* = 0.00) in Mexican Hispanics. Moreover, advanced-stage was associated with ER (PR = 0.85, 95% CI: 0.74–0.97, *p* = 0.016), PR (PR = 0.85, 95% CI: 0.73–0.99, *p* = 0.042), or HR positivity (PR = 0.86, 95% CI: 0.76–0.98, *p* = 0.028) in Mexican Hispanics. In contrast, menopausal status was noted to have an association with tumor subtypes only among non-Mexican Hispanics. Specifically, postmenopausal non-Mexican Hispanic women were less likely to have ER (PR = 0.77, 95% CI: 0.66–0.90, *p* = 0.001), PR (PR = 0.69, 95% CI: 0.55–0.86, *p* = 0.001), or HR positivity (PR = 0.77, 95% CI: 0.66–0.90, *p* = 0.001) and more likely to have TNBC (PR = 2.17, 95% CI: 1.07–4.39, *p* = 0.031).

## 4. Discussion

In this multi-institutional study focusing on Hispanic women with breast cancer, the prevalence of metabolic comorbidities was significant. Nearly 73% of the individuals included in this analysis had at least one comorbidity, and over 20% presented with two or more comorbidities—both metrics being notably higher than the national metric [8]. Overall, HR-positive breast cancer was more prevalent in non-Mexican Hispanics. Moreover, several trends with metabolic comorbidities were noted based on ethnic origin, including a higher prevalence of DM in Mexican and hyperlipidemia in non-Mexican Hispanics. Premenopausal women of Mexican origin were more likely to have an association with TNBC, while in non-Mexican Hispanic premenopausal women, there was an association noted with lobular or mixed carcinoma and ER-positive and PR-positive breast cancer. This variation among Hispanic premenopausal women of various ethnic origins is unclear. However, age at diagnosis and breast cancer prognosis have been related to breast cancer subtypes [9]. Age-dependent association between TNBC and socioeconomic and racial groups was also reported [10]. Further research to identify further underlying determinants of health among Hispanic patients of different ethnic origins might be of interest. Nevertheless, our findings suggest underlying prognosis differences between Hispanic subtypes, which remain to be confirmed in future studies.

Obesity is likely one of the most known modifiable risk factors associated with breast cancer [11]. US-born Hispanic women have a higher risk of breast cancer than their foreign-born counterparts, which increases successively in the generations living in the United States [12]. In the US, five-year breast cancer survival rates are the lowest in African-Americans (78.9%), followed by Hispanics (87%) [13]. Notably, these groups also have a higher prevalence of general [14] and central [15] obesity. In this analysis, nearly half of our overall population had obesity, with an increased majority in patients of Mexican Hispanic and Puerto Rican ethnicity. These findings suggest that the effect of obesity on breast cancer risk may also be different among the different Hispanic ethnicities, and tailored preventive efforts to reduce and educate the individuals about these possible effects are needed.

DM was another comorbid condition of interest. The Hispanic population in the United States has been reported to have higher rates of DM in both adults (80% higher than non-Hispanic whites) and children [16]. Several previous studies, including a study from Uruguay, have reported a higher risk for breast cancer among women with DM, with the elevated risk being limited to postmenopausal women [17,18,19]. Gunter et al. suggested that the glucose level moderates the risk for breast cancer more than a diagnosis of DM per se [20]. This analysis indicates that 25.6% of Hispanic women evaluated had DM, with a high prevalence found predominantly in patients of Mexican ethnicity, specifically in postmenopausal women. These findings again emphasize the importance of lifestyle modifications, including dietary control, treating DM, and improving physical activity, in this patient population to potentially improve the overall outcome of or prevent breast cancer.

HTN has also been seen to increase the risk of breast cancer [21]. In this analysis, our data suggest a higher association between HTN and HR-positive breast cancer in pre- and postmenopausal patients. These findings are consistent with Dyer et al.’s seminal study implicating HTN as a contributing factor in breast cancer development [22]. Several other studies have reported a possible link between HTN and antihypertensive drugs, especially diuretics and cancer [23,24]. The collective data suggest that HTN is another potential risk condition to address in caring for breast cancer in Hispanic and high-risk populations.

The effects of hyperlipidemia on the risk of developing breast cancer are less clear. While some studies have suggested that hyperlipidemia might increase the risk [25], Touvier et al. reported the lack of such an association in a meta-analysis suggesting a modest but statistically significant inverse association between hyperlipidemia and the risk of breast cancer [26]. Further studies are therefore required to explore the associations of hyperlipidemia with breast cancer.

Our study suggested an increased association of HR-positive breast cancer with metabolic syndrome. This syndrome is characterized by a state of insulin resistance/hyperinsulinemia and subacute chronic inflammation, and both conditions offer a plausible mechanistic link to breast cancer. Thus, in addition to their increased risk of cardiovascular morbidity and mortality, women with this syndrome represent a group at elevated risk of developing breast cancer with a possibly poorer prognosis [27]. More recently, a large study based on the Women’s Health Initiative [28] examined the association of metabolic syndrome using baseline measurements of blood glucose, triglyceride, high-density lipoprotein (HDL) cholesterol, blood pressure, waist circumference, and BMI (normal, overweight, obese) with the risk of postmenopausal breast cancer in a prospective analysis of a cohort of postmenopausal women (*n*∼21,000). These findings suggest that screening for and preventing metabolic syndrome through lifestyle changes may confer protection against breast cancer [29].

Our data analyzed the potential heterogeneity of different Hispanic ethnicities and breast cancer subtypes. It has been recognized that breast cancer subtypes might vary by race and ethnicity [30]. Our findings confirmed that, in the overall population, the most common cancer subtype in Hispanic women is HR+/HER2− [30]. On further stratification, however, non-Mexican Hispanic ethnicities were seen to have a stronger association with HR−positive breast cancers, including pre-menopausal non-Mexican Hispanic women. In contrast, pre-menopausal Mexican women had a higher association of being diagnosed with TNBC. These findings are of prognostic importance as breast cancer survival is known to vary by tumor subtype, with the highest five-year relative survival reported for patients with HR+/HER2− (92%), followed by HR+/HER2+ (89%), HR−/HER2+ (83%), and TN (77%) [30]. In addition, these findings might help in considering informed decisions for women considering preventive strategies [31,32] or screening methods [33].

The strengths of our study include the focus on a minority population and it being one of the few studies to determine the correlation of the combined metabolic comorbidities with breast cancer in this unique population. Additionally, it identifies the benefit of not grouping all Hispanics together, as we have specified several distinctive features in comorbidities and tumor characteristics between Mexican Hispanics and those of other origins. However, this study had several limitations. First, being a retrospective analysis, it did not lend itself to applying the specific metabolic syndrome criteria due to the limitations of the measures available in the archived data. For example, we used BMI as our marker for obesity, which reflects general mass-to-height and might not correspond with fat distribution measurements for abdominal obesity, hip, waist circumference, and waist-to-hip ratio. Moreover, we did not include detailed information about the subtypes of dyslipidemia due to limitations in the database. In addition, survival data were not available to confirm the implications of various associations on prognosis and overall survival.

## 5. Conclusions

This multi-institutional analysis suggests differences in tumor characteristics and metabolic abnormalities between Hispanic patient populations based on their ethnicities. While HR-positive breast cancers were more common in non-Mexicans, metabolic abnormalities were more prevalent in the Mexican population. Moreover, the metabolic conditions analyzed were noted to have strong associations with breast cancer in Mexicans, while obesity was reported to have a strong association among non-Mexicans. Given this significant prevalence of metabolic risk factors and heterogenicity of presentation among the Hispanic population with breast cancer, it would be desirable to evaluate these conditions further as part of additional efforts to decrease cancer disparities. Although our findings need to be confirmed in future, more extensive studies, increasing awareness in Hispanic patients with breast cancer about metabolic comorbidities, including obesity, would be a reasonable first step towards providing more individualized breast cancer care. The heterogeneity of the Hispanic patient population based on ethnicity is noteworthy and should be considered when designing research and generating conclusions.

## Figures and Tables

**Table 1 cancers-14-03411-t001:** Distribution of patient characteristics, overall and by Hispanic subgroups.

	Overall Sample	Mexican Hispanics	Non-Mexican Hispanics	*p*-Value
* **N** *	1020	851 (83.43%)	169 (16.56%)	
Age (mean (SD))	56.0 (11.9)	56.1 (11.9)	55.7 (11.7)	0.67
BMI (mean (SD))	30.4 (6.2)	30.9 (6.2)	28.0 (5.8)	<0.001
Premenopausal women	345 (33.8%)	280 (32.9%)	65 (38.5%)	0.16
Diagnoses				<0.001
Ductal	858 (84.4%)	734 (86.6%)	124 (73.4%)	
Lobular	76 (7.5%)	53 (6.3%)	23 (13.6%)	
Ductal and lobular (mixed)	20 (2.0%)	9 (1.1%)	11 (6.5%)	
Other	63 (6.2%)	52 (6.1%)	11 (6.5%)	
Stage of tumor				0.18
Stage I/II (early)	497 (48.7%)	420 (49.4%)	77 (45.6%)	
Stage III/IV (advanced)	278 (27.3%)	236 (27.7%)	42 (24.9%)	
Unknown	245 (24.0%)	195 (22.9%)	50 (29.6%)	
Type of surgery				<0.001
None	92 (9.1%)	62 (7.4%)	30 (17.8%)	
Lumpectomy	509 (50.3%)	455 (54.0%)	54 (32.0%)	
Mastectomy	400 (39.6%)	321 (38.1%)	79 (46.7%)	
Unknown	10 (1.0%)	4 (0.5%)	6 (3.6%)	
Receptor subtype				
ER+	696 (69.9%)	567 (68.6%)	129 (76.3%)	0.045
PR+	589 (59.2%)	482 (58.4%)	107 (63.3%)	0.23
HER2+	160 (18.2%)	129 (18.1%)	31 (18.3%)	0.95
HR+	707 (69.3%)	578 (67.9%)	129 (76.3%)	0.030
TNBC	192 (21.8%)	158 (22.2%)	34 (20.1%)	0.55
Comorbidities				
Diabetes mellitus	261 (25.6%)	237 (27.8%)	24 (14.2%)	<0.001
Hypertension	398 (39.0%)	322 (37.8%)	76 (45.0%)	0.083
Hyperlipidemia	298 (29.2%)	212 (24.9%)	86 (50.9%)	<0.001
Obesity	489 (47.9%)	434 (51.0%)	55 (32.5%)	<0.001
Coronary artery disease	43 (4.2%)	36 (4.2%)	7 (4.1%)	0.96
Metabolic abnormalities	212 (20.8%)	184 (21.6%)	28 (16.6%)	0.14
Any comorbidities	742 (72.7%)	614 (72.2%)	128 (75.7%)	0.34
Number of comorbidities				0.083
0	278 (27.3%)	237 (27.8%)	41 (24.3%)	
1	324 (31.8%)	271 (31.8%)	53 (31.4%)	
2	206 (20.2%)	159 (18.7%)	47 (27.8%)	
3	138 (13.5%)	120 (14.1%)	18 (10.7%)	
4	74 (7.3%)	64 (7.5%)	10 (5.9%)	
Hispanic ethnicities/races				NA
Mexican	851 (83.4%)	851 (100.0%)	0 (0.0%)	
Puerto Rican	5 (0.5%)	0 (0.0%)	5 (3.0%)	
Cuban	11 (1.1%)	0 (0.0%)	11 (6.5%)	
South/Central American	21 (2.1%)	0 (0.0%)	21 (12.4%)	
Spanish NOS/Hispanic NOS	129 (12.6%)	0 (0.0%)	129 (76.3%)	
Dominican Republic	3 (0.3%)	0 (0.0%)	3 (1.8%)	

ER, estrogen receptor; PR, progesterone receptor; HR, hormone receptor; TNBC, triple-negative breast cancer; HER2, human epidermal receptor 2 neu; SD, standard deviation; NOS, not otherwise specified; NA, not applicable.

**Table 2 cancers-14-03411-t002:** Association of Hispanic origin with breast cancer subtypes (ER, PR, HR, and triple negative).

	ER+ *				PR+ *			
	PR **	95% CI		*p*-Value	PR **	95% CI		*p*-Value
Non-Mexicans	1.12	1.01	1.25	0.036	1.05	0.91	1.21	0.482
Diabetes mellitus	1.04	0.94	1.15	0.418	1.02	0.89	1.16	0.804
Hypertension	1.09	1.00	1.20	0.06	1.13	1.01	1.27	0.036
Obesity	1.06	0.97	1.15	0.201	1.06	0.96	1.18	0.263
Dyslipidemia	0.97	0.88	1.07	0.515	1.01	0.89	1.14	0.884
Postmenopausal	0.99	0.9	1.08	0.796	0.9	0.8	1	0.056
Diagnosis-IDC (reference)								
Lobular	1.33	1.22	1.46	<0.001	1.35	1.17	1.55	<0.001
Lobular and ductal	1.15	0.91	1.44	0.235	1.19	0.88	1.62	0.268
Other	1.10	0.94	1.30	0.236	1.08	0.86	1.35	0.499
Stage—Early stage (reference)								
Advanced stage	0.86	0.77	0.97	0.01	0.87	0.76	1.00	0.053
Stage unknown	1.02	0.93	1.12	0.705	1.00	0.89	1.13	0.966
Surgery—None (reference)								
Lumpectomy	1.16	0.97	1.38	0.102	1.15	0.92	1.44	0.217
Mastectomy	1.07	0.9	1.28	0.436	1.09	0.87	1.36	0.439
Unknown	1.01	0.6	1.71	0.972	1.25	0.73	2.13	0.413
	**HR+ ***				**TN ***			
Non-Mexicans	1.13	1.01	1.25	0.028	0.82	0.57	1.19	0.298
Diabetes mellitus	1.01	0.92	1.12	0.788	0.82	0.57	1.17	0.268
Hypertension	1.13	1.03	1.24	0.008	0.90	0.67	1.21	0.5
Obesity	1.05	0.97	1.14	0.226	1.01	0.78	1.31	0.939
Dyslipidemia	0.98	0.89	1.08	0.703	1.24	0.9	1.7	0.185
Postmenopausal	0.97	0.88	1.06	0.488	0.89	0.68	1.16	0.377
Diagnosis-IDC (reference)								
Lobular	1.28	1.16	1.41	<0.001	0.30	0.13	0.70	0.005
Lobular and ductal	1.08	0.85	1.38	0.533	0.79	0.27	2.33	0.67
Other	1.00	0.84	1.2	0.982	0.78	0.44	1.37	0.381
Stage—Early stage (reference)								
Advanced stage	0.87	0.78	0.98	0.019	1.08	0.8	1.46	0.619
Unknown	0.99	0.90	1.09	0.768	1.1	0.8	1.52	0.549
Surgery—None (reference)								
Lumpectomy	1.11	0.94	1.32	0.221	0.57	0.38	0.85	0.006
Mastectomy	1.03	0.87	1.22	0.72	0.61	0.42	0.89	0.011
Unknown	0.89	0.52	1.54	0.68	0.78	0.22	2.76	0.702

* ER, estrogen receptor; PR, progesterone receptor; HR, hormone receptor; TN, triple-negative breast cancer; ** PR, prevalence ratio.

**Table 3 cancers-14-03411-t003:** Adjusted association of comorbidities with breast cancer subtypes (ER, PR, HR, and triple negative).

		ER+ *				PR+ *			
		PR **	95% CI		*p*-Value	PR **	95% CI		*p*-Value
Model 1	Any comorbidities	1.03	0.94	1.14	0.498	1.03	0.91	1.16	0.669
Model 2	Metabolic syndrome	1.14	1.04	1.25	0.003	1.23	1.10	1.39	<0.001
Model 3	One comorbidity	1.00	0.9	1.13	0.934	0.98	0.85	1.12	0.745
	Two	0.99	0.87	1.12	0.857	0.95	0.81	1.12	0.571
	Three	1.09	0.96	1.25	0.188	1.17	0.99	1.38	0.066
	Four	1.23	1.08	1.4	0.002	1.27	1.06	1.52	0.008
		**HR+ ***				**TNBC ***			
Model 1	Any comorbidities	1.06	0.96	1.17	0.275	1.08	0.8	1.44	0.625
Model 2	Metabolic syndrome	1.15	1.05	1.26	0.003	0.86	0.62	1.21	0.402
Model 3	One comorbidity	1.02	0.91	1.14	0.736	1.08	0.78	1.51	0.64
	Two	1.02	0.90	1.16	0.72	1.19	0.83	1.71	0.352
	Three	1.12	0.98	1.29	0.086	1.10	0.71	1.70	0.678
	Four	1.24	1.08	1.42	0.003	0.68	0.35	1.31	0.248

* ER, estrogen receptor; PR, progesterone receptor; HR, hormone receptor; TNBC, triple-negative breast cancer; CI, confidence interval; IDC, invasive ductal carcinoma; ** PR, prevalence ratio.

## Data Availability

Due to confidentiality agreements, Appendix A can only be made available to bonafide researchers subject to a non-disclosure agreement. The corresponding authors can be contacted for details of the data and how to request access.

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
