# Peer review of "Disparities in Metabolic Conditions and Cancer Characteristics among Hispanic Women with Breast Cancer: A Multi-Institutional Study"

_cancers, 2022, doi:10.3390/cancers14143411_

Round 1

Reviewer 1 Report

This manuscript describes differences in tumour characteristics and their associations with metabolic conditions among Hispanic patients with breast cancer. While the associations of common metabolic conditions with ethnicity have already been described, the disparity among Hispanic individuals based on country of origin is understudied. This study supports a more focused approach to addressing obesity and other metabolic conditions in patients, especially in the Hispanic population with breast cancer. The authors show that the heterogeneity of the Hispanic patient population based on ethnicity is noteworthy and should be taken into consideration when designing research and generating conclusions. This manuscript reviews 31 articles regarding this topic. The topic of this manuscript is up to date, interesting and well suited for your journal. The manuscript is well written and divided into 5 parts, the text is clear and easy to read. For better visualisation authors used 3 tables. I suggest checking for some spelling mistakes and grammar errors. Otherwise, I have no major concerns about this manuscript and I recommend it for publication.

Author Response

Many thanks for recognizing our efforts in this study. We have thoroughly reviewed our manuscript and fixed all the typos/grammatical mistakes in the revised manuscript. We have also extensively revised the wording of the manuscript to address all the reviewers’ comments.

Reviewer 2 Report

Thank you for proposing me to review this work. The title is very clear, the objectives are interesting and relevant and the sample is sufficient. As they say, the authors aim to increase awareness regarding the prevalence of common metabolic conditions in the Hispanic population to improve overall health and breast cancer care, including prioritizing lifestyle modifications for Hispanics and other minorities. Here are some comments in case you find them useful.

Main comments

-          I think the authors are ambitious in trying to analyse the non-Mexican hispanic population in terms of their countries or regions of origin, but they have a problem with the sample, as there are too few representatives from some countries, such as the Dominican Republic or Puerto Rico. It would be more appropriate to simply separate between MH and NMH, which is what the title of the article actually says. The authors should focus their objectives better. Separating non-Mexicans according to their origin makes no sense, given the lack of sample. Then, that part (Table 2) should be removed and focus on the differences between MH and NMH. Perhaps, Table 2 could be included as a supplementary table.

 The authors comment extensively on the supplementary tables in the text, I think they should incorporate the ones they consider most relevant as tables in the text, and leave the less informative ones as supplementary tables, this makes it easier to read and is more coherent. In other words, I think it would be advisable to restructure the article.

 I think the authors should elaborate on the reason for analysing premenopausal women separately, although some results differ for this reason between MH and NMH. What is the basis for the fact that there is an interaction between Hispanic origin and menopausal status? Could they have any ideas about this relationship? What is the main benefit of doing this analysis according to age?

Minor comments and typographical errors

-          Abstract:

o   26: “…among Mexican and non-Mexican Hispanics and patients of other ethnic origins.” I think all patients are MH or NMH, but there are no others from other ethnic origins

o   31: diabetes mellitus

-          Materials and Methods:

o   70: “…we supplemented any missing diagnostic and comorbidity information in the sample population using individual records.”   But, in Table 1, not all items total 1020 or 851 (for example, Diagnoses or Type of surgery). This modifies, slightly, the percentages obtained.

o   76: What does Spanish NOS/Hispanic NOS mean?

o   79: What does CAD mean? I think it would be better to write Coronary artery disease

o   84: PgR+ or PR+? Please always use the same abbreviations.

-          Results:

o   121: “Most of the patients in our analysis were of the Mexican ethnicity (83.4%), followed by those categorized as Spanish NOS/Hispanic NOS (12.6%) (Table 1). “This information corresponds to Table 2

o   132: “…(especially ER+) tumor subtype (76.3% vs 67.9%, p=0.03)”

o   138: In Supplementary Table 2, the type of surgery is significantly different between the MH and NMH

o   147: “of Mexican nationality (24.1%)” But the 61.9% of South/Central American are hyperlipidemic, more tan Mexican

o   158,159,160: “Overall, HR positivity was associated with non-MH subtype (PR 1.13, 95% CI 1.02-1.25, p=0.028), HTN (PR 1.13, 95% CI 1.01-1.27, p 0.008), lobular carcinoma (PR 1.29, 95%CI 1.17-1.42, p<0.001).” Some errors: Overall, HR positivity was associated with non-MH subtype (PR 1.13, 95% CI 1.01-1.25, p=0.028), HTN (PR 1.13, 95% CI 1.03-1.24, p 0.008), lobular carcinoma (PR 1.28, 95% CI 1.16-1.41, p<0.001) and early-stage cancers…

o   224: Is really 10.9% the obesity national level for the general population?

References

-          224: references 12 and 13 are cited for obesity, but are more appropriate for DM.

Author Response

Reviewer's Comments: Thank you for proposing me to review this work. The title is very clear, the objectives are interesting and relevant, and the sample is sufficient. As they say, the authors aim to increase awareness regarding the prevalence of common metabolic conditions in the Hispanic population to improve overall health and breast cancer care, including prioritizing lifestyle modifications for Hispanics and other minorities. Here are some comments in case you find them useful.

Response: Thank you for your insightful suggestions and for appreciating our work and efforts. We have reviewed and reworded the manuscript extensively to address the reviewers’ comments.

Main comments

I think the authors are ambitious in trying to analyze the non-Mexican Hispanic population in terms of their countries or regions of origin, but they have a problem with the sample, as there are too few representatives from some countries, such as the Dominican Republic or Puerto Rico. It would be more appropriate to simply separate between MH and NMH, which is what the title of the article says. The authors should focus their objectives better. Separating non-Mexicans according to their origin makes no sense, given the lack of sample. Then, that part (Table 2) should be removed and focus on the differences between MH and NMH. Perhaps, Table 2 could be included as a supplementary table.

Response: We agree entirely with your suggestions. Our sample size is insufficient to assess the heterogeneity among different non-Mexican Hispanics. Therefore, we had only presented preliminary results between non-Mexican Hispanics and Mexican Hispanics, as acknowledged in our title. In Table 2, we only presented the (unadjusted) distribution of patient characteristics by different regions of origin. We have now moved this table to the Supplementary Tables document per the suggestion.  Please see revised Supplementary Table 3.

The authors comment extensively on the supplementary tables in the text, I think they should incorporate the ones they consider most relevant as tables in the text and leave the less informative ones as supplementary tables, this makes it easier to read and is more coherent. In other words, I think it would be advisable to restructure the article.

Response: Thank you for your excellent suggestions. Based on your recommendations, we have removed Supplementary Tables 5 and 6 from the manuscript and moved Table 2 into the Supplementary Tables document as Supplementary Table 3, and considered Supplementary Table 4 as Table 3 in the revised manuscript. We think restructuring this way, tables and results read well.

I think the authors should elaborate on the reason for analyzing premenopausal women separately, although some results differ for this reason between MH and NMH. What is the basis for the fact that there is an interaction between Hispanic origin and menopausal status? Could they have any ideas about this relationship? What is the main benefit of doing this analysis according to age?

Response: Hormone affects some breast cancers and strongly depends on the age at diagnosis. Despite having a similar age at diagnosis of breast cancer among Hispanic populations, our data showed a strong interaction between age and Hispanic origin, suggesting that breast cancer subtypes may have a differential prevalence in different Hispanic populations depending on age at diagnosis.  Moreover, age at diagnosis and breast cancer prognosis depends on breast cancer subtypes (Patridge et al., JCO, 2016). Another study also reported an age-dependent association between race and triple-negative breast cancer (Linnenbringer et al., BCRT, 2020).  These findings will help in understanding prognosis differences between Hispanic subtypes. We have expanded the discussion section to provide interpretations of these data in our revised manuscript.

Minor comments and typographical errors

Abstract:

26: “…among Mexican and non-Mexican Hispanics and patients of other ethnic origins.” I think all patients are MH or NMH, but there are no others from other ethnic origins

Response: Thank you, this error has been corrected, and the statement about ‘other ethnic origins’ has been removed.

31: diabetes mellitus

Response: Thank you, this spelling has been corrected.

Materials and Methods:

70: “…we supplemented any missing diagnostic and comorbidity information in the sample population using individual records.”   But, in Table 1, not all items total 1020 or 851 (for example, Diagnoses or Type of surgery). This modifies, slightly, the percentages obtained.

Response: Thank you. We clarified in the methods section that percentages are computed among available data.  In addition, we have removed this statement from the revised manuscript. 

76: What does Spanish NOS/Hispanic NOS mean?

Response: The abbreviation ‘NOS’ has been expanded in the text. 

79: What does CAD mean? I think it would be better to write coronary artery disease.

Response: The abbreviation ‘CAD’ has been added to the text.

84: PgR+ or PR+? Please always use the same abbreviations.

Response: This error has been corrected to use a consistent abbreviation throughout the body of the manuscript.

Results:

121: “Most of the patients in our analysis were of the Mexican ethnicity (83.4%), followed by those categorized as Spanish NOS/Hispanic NOS (12.6%) (Table 1). “This information corresponds to Table 2.

Response: This error has been corrected, and the correct table has been cited.

132: “…(especially ER+) tumor subtype (76.3% vs 67.9%, p=0.03)”

Response: Thank you for highlighting this error. This has been corrected.

138: In Supplementary Table 2, the type of surgery significantly differs between the MH and NMH.

Response: The statement has been reworded to reflect these findings better.

147: “of Mexican nationality (24.1%)” But the 61.9% of South/Central Americans are hyperlipidemic, more than Mexican

Response: The statement has been reworded to reflect our findings correctly.

158,159,160: “Overall, HR positivity was associated with non-MH subtype (PR 1.13, 95% CI 1.02-1.25, p=0.028), HTN (PR 1.13, 95% CI 1.01-1.27, p 0.008), lobular carcinoma (PR 1.29, 95%CI 1.17-1.42, p<0.001).” Some errors: Overall, HR positivity was associated with non-MH subtype (PR 1.13, 95% CI 1.01-1.25, p=0.028), HTN (PR 1.13, 95% CI 1.03-1.24, p 0.008), lobular carcinoma (PR 1.28, 95% CI 1.16-1.41, p<0.001) and early-stage cancers…

Response: Thank you for your comment. The appropriate changes have been made in the text.

224: Is really 10.9% the obesity national level for the general population?

Response: This statement and citation have been removed from the body of the manuscript.

References

224: references 12 and 13 are cited for obesity, but are more appropriate for DM.

Response: Thank you for your comment. These references have been removed from the manuscript.

Reviewer 3 Report

The authors present an interesting study assessing the correlation between metabolic abnormalities and breast cancer in Mexican and non-Mexican Hispanic women. There have been a few articles in the recent years that have discussed the incidence of breast cancer in Hispanic women.

https://www.nature.com/articles/s41467-021-22478-5

However, a core strength of this manuscript is to assess the underlying metabolic conditions.

Overall, I think the article is well written and I recommend acceptance with some minor changes. In Table 1 and through the text, the authors mention “metabolic abnormalities”. It would be important to describe what these abnormalities are.

Author Response

Thank you for your appreciation and valuable comments. The metabolic abnormalities have been mentioned in the introduction section to highlight the included variables in our study. In addition, the manuscript has been revised and reworded extensively to address the reviewers' comments.

Round 2

Reviewer 2 Report

Thank you for your response

As you have made the changes I proposed, I think your article can be read more easily and comprehensibly. I have not been able to check whether all the changes have been introduced (36-37, 84) and I believe that in row 30-31 there is an incomplete sentence, but surely a final reading by the authors will correct any possible errors. I think it is an article that adds new information to the current knowledge of this subject. Best regards